

# Tomographic reconstruction of atmospheric gravity wave parameters from airglow observations

Rui Song [1,2], Martin Kaufmann [1], Jörn Ungermann [1], Manfred Ern [1], Guang Liu [3], and Martin Riese [1]

[1]Institute of Energy and Climate Research, Stratosphere (IEK-7), Research Centre Jülich, 52425 Jülich, Germany
[2]Physics Department, University of Wuppertal, 42097 Wuppertal, Germany
[3]Key Laboratory of Digital Earth Sciences, Institute of Remote Sensing and Digital Earth, Chinese Academy of Sciences, Beijing, China

*Correspondence to:* R. Song (r.song@fz-juelich.de)

**Abstract.**

Gravity waves (GWs) play an important role in atmospheric dynamics. Especially in the mesosphere and lower thermosphere (MLT) dissipating GWs provide a major contribution to the driving of the global wind system. Therefore global observations of GWs in the MLT region are of particular interest. The small scales of GWs, however, pose a major problem for the observation

of GWs from space. We propose a new observation strategy for GWs in the mesopause region by combining limb and sub-limb satellite-borne remote sensing measurements for improving the spatial resolution of temperatures that are retrieved from atmospheric soundings. In our study, we simulate satellite observations of the rotational structure of the $O_2$ A-band nightglow. A key element of the new method is the ability of the instrument or the satellite to operate in so called 'target mode', i.e. to stare at a particular point in the atmosphere and collect radiances at different viewing angles. These multi-angle measurements

of a selected region allow for tomographic reconstruction of a 2-dimensional atmospheric state, in particular of gravity wave structures. As no real data is available, the feasibility of this tomographic retrieval is carried out with simulation data in this work. It shows that one major advantage of this observation strategy is that much smaller scale GWs can be observed. We derive a GW sensitivity function, and it is shown that 'target mode' observations are able to capture GWs with horizontal wavelengths as short as ∼50 km for a large range of vertical wavelengths. This is far better than the horizontal wavelength

limit of 100-200 km obtained for conventional limb sounding.

## 1 Introduction

Miniaturization in remote sensing instrumentation as well as spacecraft technology allows for the implementation of highly focused satellite missions, for example to observe airglow layers in the mesosphere and lower thermosphere (MLT) region. The MLT extends between about 50 and 110 km in the Earth's atmosphere, and is highly affected by atmospheric waves,

including planetary waves, tides and gravity waves (GWs) which are mainly excited in the lower atmosphere (Vincent, 2015). Atmospheric GWs are the main driver for the large-scale circulation in the MLT region with considerable effects on the atmospheric state and temperature structure (Garcia and Solomon, 1985; Holton, 1982; Lindzen, 1981).



Temperature is a key quantity to describe the atmospheric state, and it is a valuable indicator to identify and quantify atmospheric waves, such as GWs (e.g., Fritts and Alexander 2003, and reference therein). As GWs displace airparcels adiabatically both in vertical and horizontal directions, this process affects the temperature of the atmosphere. Assuming linear wave theory, GW-related fluctuations in different parameters (wind, temperature, density, etc.) are directly connected via the linear polar-

ization relations (e.g., Fritts and Alexander 2003). Therefore, amplitudes, wavelengths, and phases of a GW can be determined from its temperature structure (Fritts et al., 2014; Ern et al., 2004, 2017).

Over the last few decades, data from satellite and aircraft instruments have been extensively used to characterize vertically resolved gravity wave parameters. Utilizing limb soundings, they include temperature or density data acquired by the Limb Infrared Monitor of the Stratosphere (LIMS) (Fetzer and Gille, 1994, 1996), the Global Positioning System (GPS) radio occulta-

tion (RO) (Tsuda et al., 2000), the Microwave Limb Sounder (MLS) (Wu and Waters, 1996), Cryogenic Infrared Spectrometers and Telescopes for the Atmosphere (CRISTA) (Eckermann and Preusse, 1999), Sounding of the Atmosphere using Broadband Emission Radiometry (SABER) (Ern et al., 2011; Preusse et al., 2009a), High Resolution Dynamics Limb Sounder (HIRDLS) (Alexander et al., 2008) and Gimballed Limb Observer for Radiance Imaging of the Atmosphere (GLORIA) aircraft (Riese et al., 2014; Kaufmann et al., 2015; Ungermann et al., 2010b, 2011). Utilizing nadir sounding observations, they include data

by the Atmospheric Infrared Sounder (AIRS) (Alexander and Barnet, 2007; Hoffmann and Alexander, 2009; Ern et al., 2017) and the Advanced Microwave Sounding Unit (AMSU) (Wu, 2004).

Typical limb sounders provide middle atmosphere temperature data with a vertical resolution of 1-3 km assuming a horizontally homogeneous atmosphere. In most cases, vertical structures of small horizontal-scale GWs are characterized by separating a background temperature profile from the measured profiles. To this background temperature, the average temperature structure

of the atmosphere contributes, as well as several different modes of planetary waves (e.g., Ern et al. 2009), and tides (e.g., Forbes et al. 2006). The final results of this procedure are altitude profiles of temperature perturbations due to GWs. Temperature data obtained from limb sounding instruments exhibit a very good vertical resolution, but suffer from a poor horizontal resolution along the instrument's line-of-sight, thus limiting the visibility of short horizontal waves. Ern et al. (2004) proposed to combine the phases provided by the wave analysis of adjacent temperature vertical profiles to estimate the horizontal wavelength of

GWs. This approach has been successfully applied to retrieve GWs with vertical wavelengths between 6 and 30 km and horizontal wavelengths larger than 100 km from CRISTA-2 measurements. This method has been used also for several other datasets (Alexander et al., 2008; Wright et al., 2010; Ern et al., 2011).

A general limitation of all methods based on limb sounding is the poor along line-of-sight resolution of this kind of measurement, which is typically a few hundred kilometers. A few existing and upcoming limb sounders try to mitigate this general

limitation by oversampling the data in horizontal direction (Carlotti et al., 2001; Livesey and Read, 2000) . The GLORIA limb sounder utilizes a tomographic reconstruction technique, which leads to a horizontal resolution of 20 km (Ungermann et al., 2010b). In this work, we present another measurement strategy to detect atmospheric structures, whose spatial dimensions are neither covered by conventional limb sounding nor satellite- or ground based nadir sounding. It is applicable to a low-cost nanosatellite utilizing a remote sensing instrument to measure atmospheric temperature and a high precision pointing system.

Simply speaking, the satellite is commanded in such a way that the instrument observes a certain volume in the atmosphere





multiple times while the satellite is flying by. This results in multi-angle observations of the target volume, such that a retrieval scheme which differs from classical limited-angle tomography can be applied.

In Sect. 2, we present the observation strategy of 'target mode' observations. Sect. 3 describes the forward modeling of such 'target mode' measurements, which is based on a 2D ray tracing, an oxygen atmospheric band (A-band) airglow emission
model, a gravity wave perturbation, and the corresponding radiative transfer. The retrieval algorithm is presented in Sect. 4. In Sect. 5, the performance of the 'target mode' tomographic retrieval is tested with simulated measurements. A sensitivity study is used to analyze its performance in deriving GW fine structures compared with pure limb tomographic retrieval. The conclusion is given in Sect. 6.

## 2  Observation strategy

The detection of small scale structures in Earth's atmosphere in 2D requires new instruments or measurement strategies, as stated above. One of those is to observe atmospheric volumes from different viewing directions, e.g. by staring at one particular region while the instrument is flying by. Such tomographic retrievals have been demonstrated and implemented in a variety of measurements for different purposes, including Carlotti et al. (2001) for Michelson Interferometer for Passive Atmospheric Sounding (MIPAS), Livesey and Read (2000) for MLS, Ungermann et al. (2010a) for Process Exploration through
Measurements of Infrared and millimetre-wave Emitted Radiation (PREMIER), and Ungermann et al. (2011); Kaufmann et al. (2015) for the airborne GLORIA instrument. In this work, we propose to combine satellite-borne limb and sub-limb measurements of a nightglow layer in such a way that we obtain multi-angle observations of a particular air volume as well. We name this observation strategy 'target mode', i.e. we adopt the same expression as for similar measurements in Earth observation. In the following, the capabilities of 'target mode' observations will be discussed for a specific sequence of satellite
pointing maneuvers.

Figure. 1 illustrates the viewing geometry of the 'target mode' observation, which incorporates limb and sub-limb sounding measurements. When the satellite is operated in 'target mode', the instrument will start to observe the target atmospheric volume by forward limb sounding first. The instrument will keep this limb view for a period of time, and multiple consecutive vertical profiles will be taken during this time. Then, the instrument will switch to a forward sub-limb view with a 24.5°
viewing angle below horizon. This viewing angle is also constant during the sub-limb observations. In this way, the volume will be scanned twice by the limb and sub-limb observations. Depending on the flexibility and possible speed of satellite operations more viewing angle positions could be used, for example another position with a viewing angle >24.5° as indicated in Fig. 1. After the satellite overpasses the target volume, the same measurement sequence will be applied by back-looking at the target volume.

Figure. 2 shows how the line-of-sights (LOSs) of the measurements overlap with each other in the orbit plane under limb sounding and 'target mode', respectively. For an assumed orbit altitude of 600 km, the corresponding measurement time for taking the measurement sequences for the limb sounding mode shown in Fig. 2 (a) is 1.6 minutes. We further assume a high measurement frequency of 10 sec per vertical profile. As can be seen from Fig. 2 (a), for the limb sounding mode the LOSs





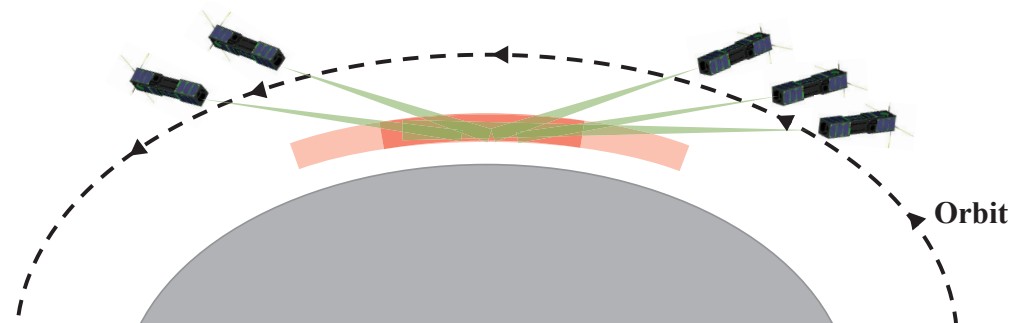

**Figure 1.** Viewing geometry of 'target mode' observations of a region in a mesospheric emission layer. This observation mode consists of forward limb, forward sub-limb and backward sub-limb measurements. The sub-limb measurements are taken with two different viewing angles, 24.5° when the satellite is far from the target region and 33° when it is relatively closer. The viewing angle is the angle relative to the instantaneous horizon, with horizon defined as Earth surface for a given but temporally changing satellite position.

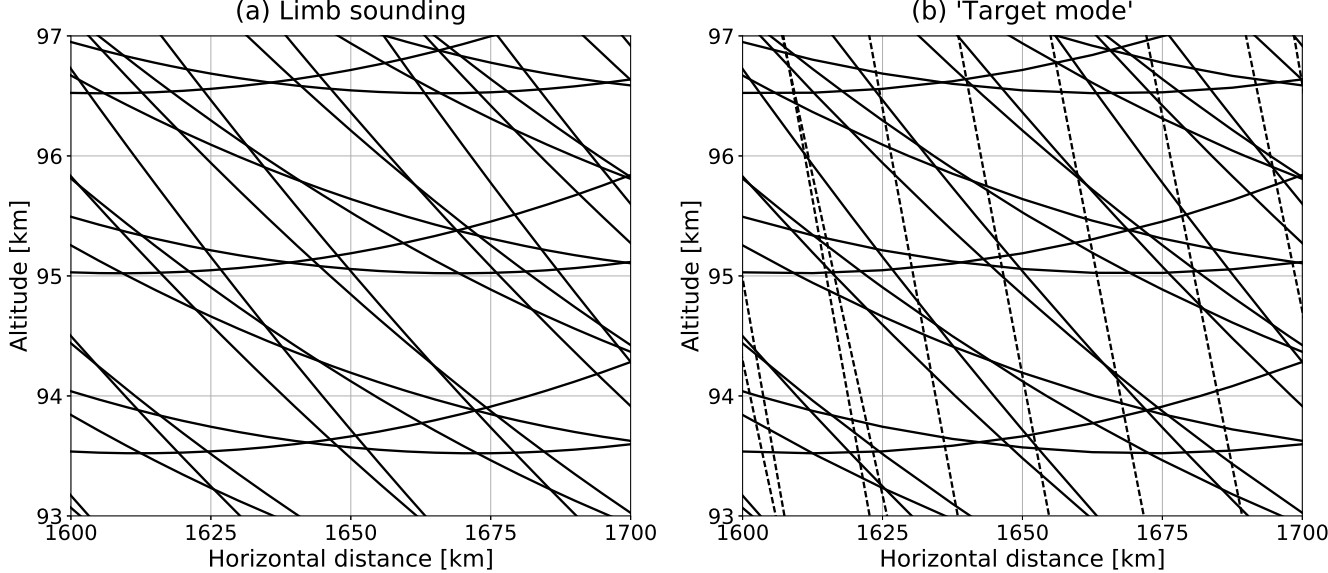

**Figure 2.** Central measurement track of an imaging instrument in the pure limb sounding and 'target mode'. The solid lines represent limb measurements and dashed lines represent sub-limb measurements. Satellite viewing geometry of Fig. 2 (b) is the same as Fig. 1.

of consecutive limb profiles overlap, which means that the same atmospheric volume is observed under different directions. While in the 'target mode' (Fig. 2 (b)), sub-limb measurements contribute further information by intersecting the observed volume at different viewing angles about 3.3 minutes after the limb observations were taken.



## 3   Forward model

### 3.1   O$_2$ A-band nightglow emission model

The observation strategy presented in the previous section requires that the observed emission is restricted to a limited altitude range and that any emissions from lower parts of the atmosphere or Earth's surface cannot reach the instrument. This requires

that the atmosphere below the emission layer needs to be optically thick for those emissions. This limits the number of potential airglow emissions significantly, because most of them are hotband transitions between two excited vibrational states. The number density of the lower state of a hotband transition is typically too low to absorb background radiation from the lower atmosphere. Therefore we have to search for airglow emissions, whose lower state is a ground state of a frequent atmospheric species. This is the case for the O$_2$ A-band nightglow emission. The emitting electronic state is excited in a two-step Barth

process (Burrage et al., 1994; Greer et al., 1981):

$$O + O + M \rightarrow O_2^* + M \tag{1}$$

where M is an O$_2$ or N$_2$ molecule, and O$_2^*$ is an excited O$_2$ molecule. Then, O$_2^*$ state is quenched to a lower electronic state O$_2(b^1\Sigma)$, which emits O$_2$ A-band radiation:

$$O_2^* + O_2 \rightarrow O_2(b^1\Sigma) + O_2 \tag{2}$$

Loss mechanisms for both O$_2(b^1\Sigma)$ and its undefined precursor O$_2^*$ include quenching by O$_2$, O$_3$, N$_2$ or O and spontaneous emission. The A-band volume emission rate (VER) $\eta$, in photons $\cdot$ s$^{-1}$ $\cdot$ cm$^{-3}$, is thus (McDade et al., 1986):

$$\eta = \frac{A_1 k_1 [O]^2 [M][O_2]}{(A_2 + k_2[O_2] + k_3[N_2] + k_4[O])(C_{O_2}[O_2] + C_O[O])} \tag{3}$$

where [] refers to the number density of the species within the brackets. $A_1$ is the A-band transmission probability, $A_2$ is the total transition probability of the zeroth vibrational level of the O$_2(b^1\Sigma)$ state (Vallance Jones, 1974). $k_1$ is the reaction

coefficient rate for reaction Eq. (1) (Campbell and Gray, 1973). The quenching rates for O$_2$, N$_2$ and O are denoted by $k_2$, $k_3$, and $k_4$, respectively. $C_{O_2}$ and $C_O$ describe quenching rates of O$_2^*$ by O$_2$ and O. All rate constants utilized in this work are taken from Sheese (2011). A typical vertical profile of a modeled O$_2$ A-band nightglow emission from 80-110 km is shown in Fig. 3. The temperature $T$ and number densities of O$_2$, N$_2$ and O are taken from the Hamburg Model of the Neutral and Ionized Atmosphere (HAMMONIA) (Schmidt et al., 2006) model run at 30° N and 88° E for 22:00 local solar time. The intensity of

the O$_2$ A-band nightglow limb emission peaks at around 93 km; typical peak values are $3 \times 10^3$ photons $\cdot$ s$^{-1}$ $\cdot$ cm$^{-3}$.

Since the lifetime of the O$_2(b^1\Sigma)$ state is more than 12 sec, it can be assumed that the molecule is in rotational local-thermodynamic equilibrium (Vallance Jones, 1974). This allows to derive the kinetic temperature of the atmosphere from the rotational band structure of the emissions. Under thermal equilibrium conditions, the O$_2$ A-band rotational excitation follows the Boltzmann distribution at a rotational temperature $T$, which is assumed to be equal to the background temperature.

The number of photons that appears in an individual rotational line is given by $\eta_{\text{rot}}$:

$$\eta_{\text{rot}} = \eta \frac{g'}{Q(T)} \exp\left(\frac{-hcE'}{kT}\right) A_1 \tag{4}$$




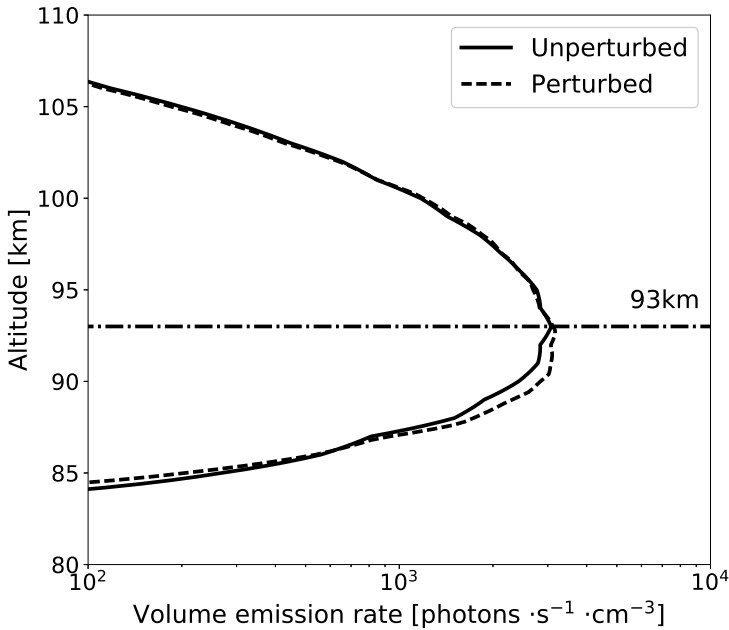

**Figure 3.** Modeled $O_2$ A-band nightglow emission profile at $30°$ N and $88°$ E for 22:00 local solar time as simulated by the HAMMONIA model. The solid curve represents an $O_2$ A-band profile for unperturbed conditions, and the dashed curve represents a profile perturbed by a GW with 15 km vertical wavelength and 5 K amplitude.

where $h$ is the Planck constant, $c$ is the light speed, $k$ is the Boltzmann constant. $E'$ and $g'$ are the upper state energy and upper state degeneracy, respectively. $A_1$ is the Einstein coefficient of the transition. $Q(T)$ is the rotational partition function:

$$Q(T) = \sum g' \exp(\frac{-hcE'}{kT})$$

(5)

A subset of six emission lines has proven to give an optimal setup for a potential satellite mission aiming to the derivation of
5  kinetic temperature from the $O_2$ A-band. These six lines show both positive and negative temperature dependence of rotational structures, also strong, medium and weak dependence are included, as shown in Fig. 4.

### 3.2 Wave perturbations

$O_2$ A-band emissions are affected by gravity waves due to the vertical displacement of constituents and temperature changes associated to the waves. Following conventional assumption, we consider an adiabatic and windless atmosphere. A monochro-
10  matic wave perturbation added in background temperature $T_0$ at position $(x, z)$ can be written as:

$$T(x, z, t) = T_0(x, z, t) + A\cos\left(\frac{2\pi x}{\lambda_x} + \frac{2\pi z}{\lambda_z} - \hat{\omega}t\right)$$

(6)





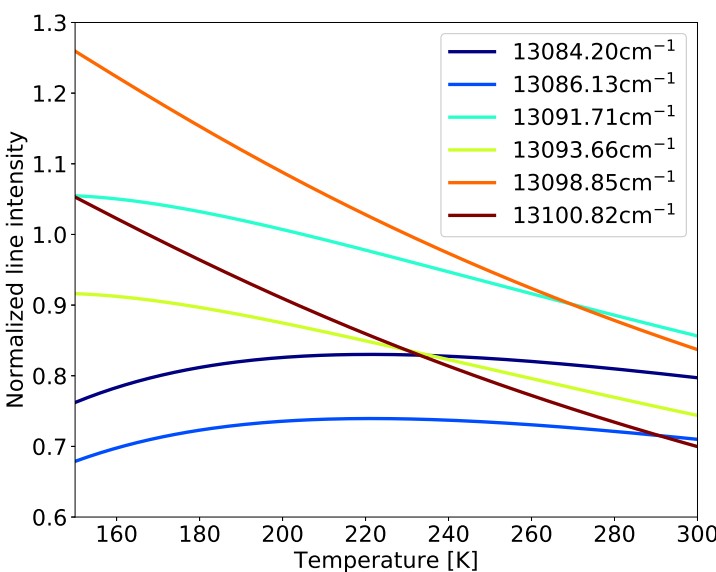

**Figure 4.** Temperature dependence of six rotational lines of the $O_2$ A-Band. The line center wavenumbers for the lines are given in the figure legend. The intensity is normalized around the maximum intensity for a temperature of 230 K.

with the wave amplitude $A$, the vertical wavelength $\lambda_z$, the horizontal wavelength $\lambda_x$ and the intrinsic frequency $\hat{\omega}$ of wave perturbation. We used the following expression (Ward, 1999) to calculate the vertical displacement $\delta z$ of an air parcel from its equilibrium height $z + \delta z$:

$$T(x, z, \delta z) \approx T(x, z) + (\Gamma_{ad} - \Gamma)\delta z \tag{7}$$

where $\Gamma$ and $\Gamma_{ad}$ are the local and adiabatic lapse rates, respectively. Then, the perturbed density (background density plus perturbation) $\rho'$ at fixed height $z$ can be calculated as density at equilibrium height $z + \delta_z$:

$$\rho'(x, z) = \rho(x, z, \delta z) \approx \rho(x, z)\exp^{-\kappa\delta z/H} \tag{8}$$

with the scale height $H$. In the quantity $\kappa = (c_p/c_v - 1)$, $c_p$ and $c_v$ represent heat capacities at constant pressure and volume, respectively. Given $\rho'$, the number densities for perturbed major gases are calculated as (Liu, 2003; Vargas et al., 2007):

$$\frac{[N_2]'}{[N_2]} = \frac{[O_2]'}{[O_2]} = \frac{\rho'}{\rho} \tag{9}$$

Because the mixing ratio of atomic oxygen is not constant with altitude, the perturbed volume mixing ratio $v'$ is calculated as follows (Ward, 1999):

$$v'_O(x, z) = v_O(x, z, \delta_z) \approx v_O(x, z + \delta_z) \tag{10}$$



Figure. 3 shows a perturbed $O_2$ A-band volume emission rate profile perturbed by a 1D GW with a vertical wavelength $\lambda_z$ of 15 km and an amplitude $A$ of 5 K.

### 3.3 Ray tracing

To model the instrument's LOS in a three dimensional atmosphere we utilize a module of the Atmospheric Radiative Transfer Simulator (ARTS) (Buehler et al., 2005). ARTS is a free open-source software program that simulates atmospheric radiative transfer. It focuses on thermal radiation from the microwave to the infrared spectral range. The second version of ARTS (Eriksson et al., 2011) allows simulations for one-, two- or three-dimensional atmospheres. In this study, the relative orientations of LOS is selected to be parallel to the orbit plane. Thus, it is assumed that the LOS is in the orbit plane, and a 2D ray tracing can be applied with ARTS-2.

### 3.4 Radiative transfer

The observed spectral irradiance $I(v)$, in $\text{photons} \cdot \text{s}^{-1} \cdot \text{cm}^{-2}$, is a path integral along the line-of-sight:

$$I(v) = \int\limits_{-\infty}^{\infty} \eta(s)_{\text{rot}}\, D(v,s)\, \exp[-\int\limits_{-s}^{\infty} n(s')\,\sigma(s')\,ds']\,ds \tag{11}$$

where $s$ is the distance along the line-of-sight, $n$ is the $O_2$ number density, $\sigma$ is the absorption cross section, and $D(v)$ is the Doppler line shape for the spectral line centered at wavenumber $v$. In our case, for altitudes above 85 km, the atmosphere is assumed to be optically thin and the self-absorption term can be omitted (Sheese, 2011).

The spectral range considered in this work is 13082-13103 $\text{cm}^{-1}$ and contains six emission lines. The central wavenumbers of these lines are given in Fig. 4. In Fig. 5, limb radiance spectra are simulated at different altitudes (86-115.5 km with 1.5 km interval). To make a trade-off between bandwidth and instrument size, a spectral resolution of 0.8 $\text{cm}^{-1}$ is chosen. The temperature dependence of the lines in this spectral interval is illustrated in Fig. 4.

### 4 Retrieval model

The tomographic retrieval presented here is similar to the widely used optimal estimation approach (Rodgers, 2004). The measurement space is represented by vector $\boldsymbol{y}$ and the unknown atmospheric state is represented by vector $\boldsymbol{x}$. The forward model $\boldsymbol{f}(\boldsymbol{x})$ provides the simulated spectrum based on a given atmospheric state $\boldsymbol{x}$:

$$\boldsymbol{y} = \boldsymbol{f}(\boldsymbol{x}) + \boldsymbol{\epsilon} \tag{12}$$

where $\boldsymbol{\epsilon}$ is the measurement error. The inversion problem of Eq. (12) is generally ill-posed, and the solution is not unique. Following Tikhonov and Arsenin (1977) and Rodgers (2004), the cost function $\boldsymbol{J}$ is complemented by a regularization term:

$$\boldsymbol{J}(\boldsymbol{x}) = (\boldsymbol{f}(\boldsymbol{x}) - \boldsymbol{y})^T \mathbf{S}_\epsilon^{-1} (\boldsymbol{f}(\boldsymbol{x}) - \boldsymbol{y}) + (\boldsymbol{x} - \boldsymbol{x_a})^T \mathbf{S}_a^{-1} (\boldsymbol{x} - \boldsymbol{x_a}) \tag{13}$$



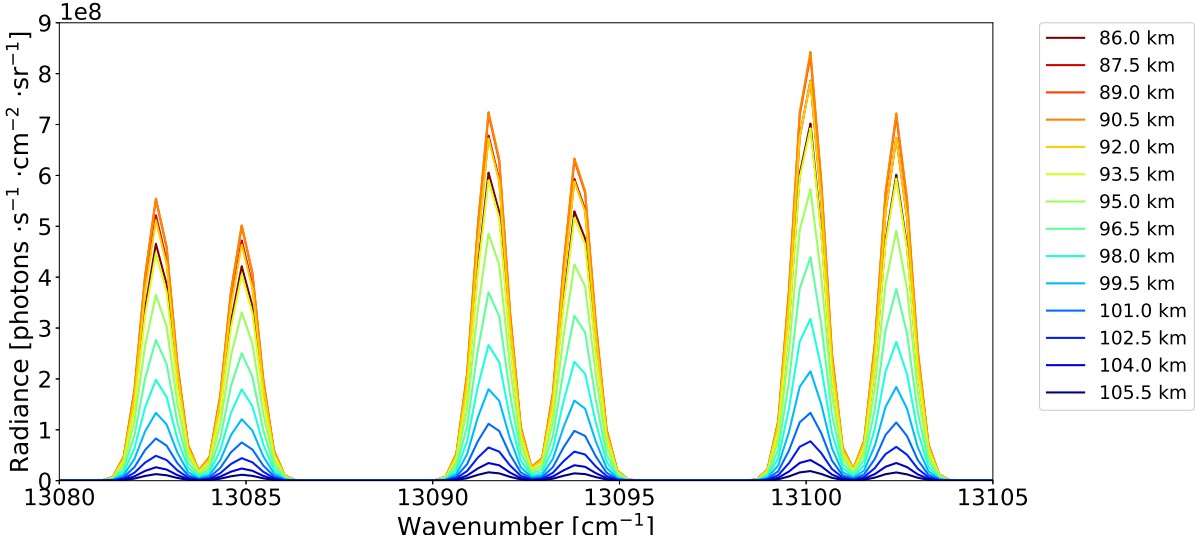

**Figure 5.** Modeled limb spectra for different tangent altitudes. The background atmosphere is taken from the HAMMONIA model, around $30°$ N and $88°$ E for 22:00 local solar time.

where matrix $\mathbf{S}_a^{-1}$ is the inverse covariance or regularization matrix, $\mathbf{S}_\epsilon$ is the covariance matrix of the measurement error, and $\boldsymbol{x_a}$ represents the a priori data. The a priori data is usually taken as the climatological mean of the retrieved quantities. It ensures that a unique and physically meaningful solution can be obtained. The inversion of the forward model (Eq. (12)) can be formulated as a minimization of the cost function $\boldsymbol{J}(\boldsymbol{x})$ given in Eq. (13). To solve the non-linear minimization, we adopt a
5   Levenberg-Marquardt iteration scheme.

### 4.1   2-dimensional regularization matrix

The design of the 2-dimensional regularization matrix $\mathbf{S}_a^{-1}$ in Eq. (13) is of considerable importance to the retrieval results. Here, we used a combination of zeroth and first order Tikhonov regularization (Tikhonov and Arsenin, 1977):

$$\mathbf{S}_a^{-1} = (\alpha_0 \mathbf{L}_0^T \mathbf{L}_0 + \alpha_1^x \mathbf{L}_1^{x\,T} \mathbf{L}_1^x + \alpha_1^y \mathbf{L}_1^{y\,T} \mathbf{L}_1^y) \tag{14}$$

10   where the weighting parameters $\alpha_0$, $\alpha_1^x$ and $\alpha_1^y$ control the overall strength of the regularization term added in Eq. (13). Large values of weighting parameters will result in an over-regularized result, while a small value will give a noisy or no solution. The parameter $\alpha_0$, $\alpha_1^x$ and $\alpha_1^y$ also balance the contribution of the zeroth and the two directional first order regularization terms. $\mathbf{L}_0$ is an identity matrix that constrains the result to the absolute value of $\boldsymbol{x_a}$. Matrix $\mathbf{L}_1$ maps $\boldsymbol{x}$ onto its first order derivative



in the vertical and horizontal directions:

$$\mathbf{L}_1^x(i,j) = \begin{cases} 1 & \text{if } j = i+1 \\ -1 & \text{if } j = i \\ 0 & \text{otherwise} \end{cases}, \qquad \mathbf{L}_1^y(i,j) = \begin{cases} 1 & \text{if } j = i+m \\ -1 & \text{if } j = i \\ 0 & \text{otherwise} \end{cases} \tag{15}$$

As we convert the 2D atmospheric temperature to a vector $x$ row by row, $\mathbf{L}_1^x$ is thus a $(l-1) \times l$ matrix with $l$ to be the number of elements in $x$. $\mathbf{L}_1^y$ is a $(l-m) \times l$ matrix with $m$ to be the number of elements contained in each row of the 2D atmospheric

volume.

## 4.2 Averaging kernel matrix.

Following the concept of Rodgers (2004), the effect of the regularization onto the retrieval result can be quantified by the averaging kernel (AVK) matrix:

$$\mathbf{A} = (\mathbf{S}_a^{-1} + \boldsymbol{f}'(\boldsymbol{x})^T \mathbf{S}_\epsilon^{-1} \boldsymbol{f}'(\boldsymbol{x}))^{-1} \boldsymbol{f}'(\boldsymbol{x})^T \mathbf{S}_\epsilon^{-1} \boldsymbol{f}'(\boldsymbol{x}) \tag{16}$$

where $\boldsymbol{f}'$ is the Jacobian matrix of the forward model $\boldsymbol{f}$ at atmospheric state $\boldsymbol{x}$. The measurement contribution vector can be obtained from the AVK by the sum over each row of $\mathbf{A}$. If the measurement contribution value is close to 1, most information of the retrieval result is determined by the measurements and not by the absolute value of the a priori data. The averaging kernel matrix can also be used to deduce the spatial resolution of retrieval result. For 1D retrievals, the vertical resolution is described by calculating the full width at half maximum (FWHM) of the corresponding row of the AVK matrix. For 2D retrievals, the

row needs first to be reshaped into two dimensions, and then the FWHM method is used to calculate the resolution along each axis (Steck et al., 2005).

## 5 Numerical experiments

### 5.1 Simulation setup

In this section, we present the experimental results of tomographic temperature retrievals using simulated 'target mode' mea-

20 surements with $1\%$ noise added. Synthetic measurements are generated by imprinting a gravity wave structure onto a smooth model atmosphere, as described in Sect. 3.2. As the amplitude is the most important feature of a GW with respect to energy, the assessment in the next step focuses on how well the wave amplitude can be reproduced from the retrieval results. The wave vector investigated in this case is assumed along the direction of line-of-sight, where the largest amplitude suppression is provided (Preusse et al., 2009b).

In our study, the spacing of the atmospheric grids is very important for both the forward and the retrieval model. To reduce the impact of the discretization on the synthetic measurements, the atmospheric grid in the forward model should be finely sampled. The atmospheric grid used in the forward model has a vertical spacing of 250 m and horizontal spacing of 5 km. In the inverse procedure, the sampling can be coarser: 500 m vertical spacing and 12.5 km horizontal spacing in our case.

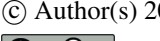



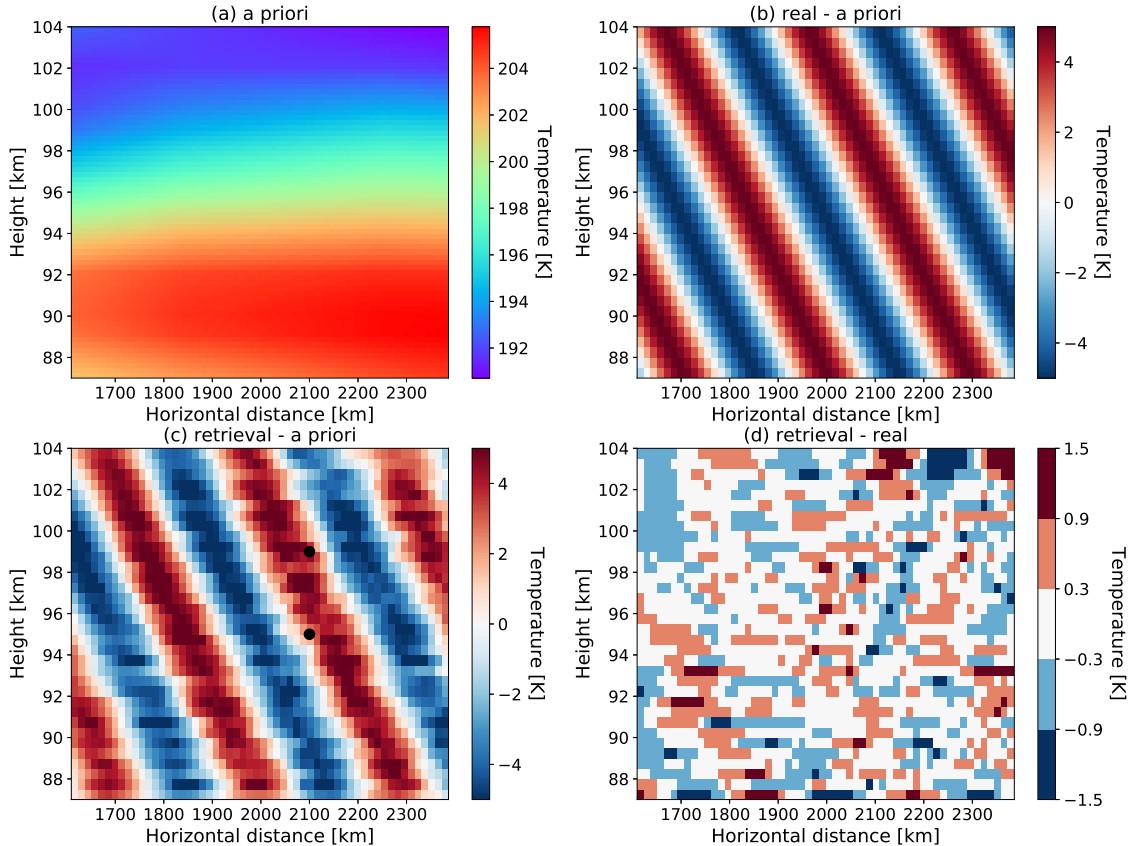

**Figure 6.** Retrieval result using simulated data. The background atmosphere is taken from the HAMMONIA run between 30° N and 36° N, 90° E, around 22:00 local solar time. The a priori atmosphere is depicted in panel (a). The difference between the perturbed atmosphere and the a priori is shown in panel (b). The retrieved wave perturbation, which is obtained by subtracting the a priori from the retrieval result, is given in panel (c). The two black dots correspond to the retrieval points selected for Fig. 7. The difference between the retrieval result and the true state of atmosphere is shown in panel (d).

## 5.2 Example of a gravity wave parameter retrieval

Figure. 6 illustrates the performance of the tomographic retrieval approach for an atmosphere disturbed by a gravity wave with a horizontal and vertical wavelength of 300 km and 15 km, respectively. The atmospheric condition as well as the sampling patterns are the same as in the previous sections. The integration time is assumed to be 10 seconds for limb measurements and 5 | 15 seconds for sub-limb measurements. The a-priori data used in this case study is depicted in Fig. 6 (a). The simulated gravity



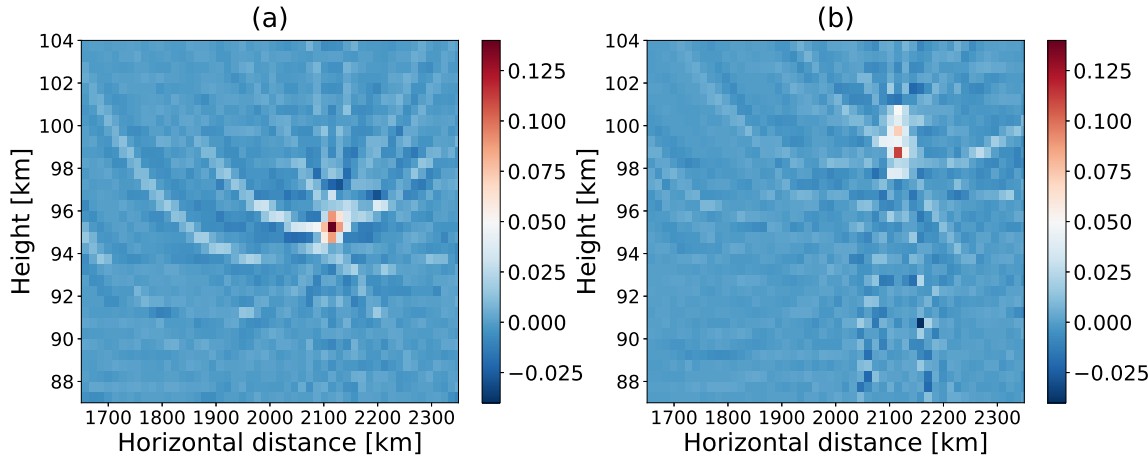

**Figure 7.** Averaging kernel matrix for two different retrieval points, allocated at the geographical position of highest values (red color). Figure (a) is for a point coinciding with the observational grid, whereas Figure (b) shows a point, whose vertical position is in between two tangent altitudes of the limb observations; for details see text.

wave has an amplitude of 5 K (Fig. 6 (b)). The retrieved temperature perturbation is shown in Fig. 6 (c). We can clearly see that the wave structure is well reproduced between 87 and 104 km height. The horizontal coverage of such a group of 'target mode' measurements is around 800 km. Fig. 6 (d) depicts the difference between simulated wave and retrieved wave, with an average error of about 0.5 K.

The spatial resolution of the retrieved data is usually described by the rows of the averaging kernel matrix, **A**. The deviation of **A** from the identity matrix gives insight into the smoothing introduced by the regularization. For example, if two adjacent grid points share one piece of information, the corresponding information content would be 0.5. By reordering a single row of **A** according to their vertical and horizontal coordinates, the influence of each point on the retrieval result can be revealed. Fig. 7 shows the 2-D averaging kernel matrix of two selected data points, which are marked as black dots in Fig. 6 (c).

Figure. 7 (a) shows the averaging kernel matrix for a point positioned at 2100 km along track and 95 km altitude. It indicates that the measurement contribution is sharply centered around this point. A minor part of the information comes from other altitudes on parabola shaped tracks, which corresponds to the line-of-sights of observations, whose tangent altitude is below 95 km or which are sub-limb observations. For Fig. 7 (b), the data point is placed at 2100 km along track and 99 km altitude. In contrast to Figure (a), this data point is not right on the tangent altitude of an observation, but in-between two observations.

Because this point is not placed exactly at one of the tangent altitudes, main contributions to the retrieved value come from measurements of adjacent grid points. According to Fig. 7 (a), a vertical resolution of 1.3 km and horizontal resolution of 35 km can be achieved.



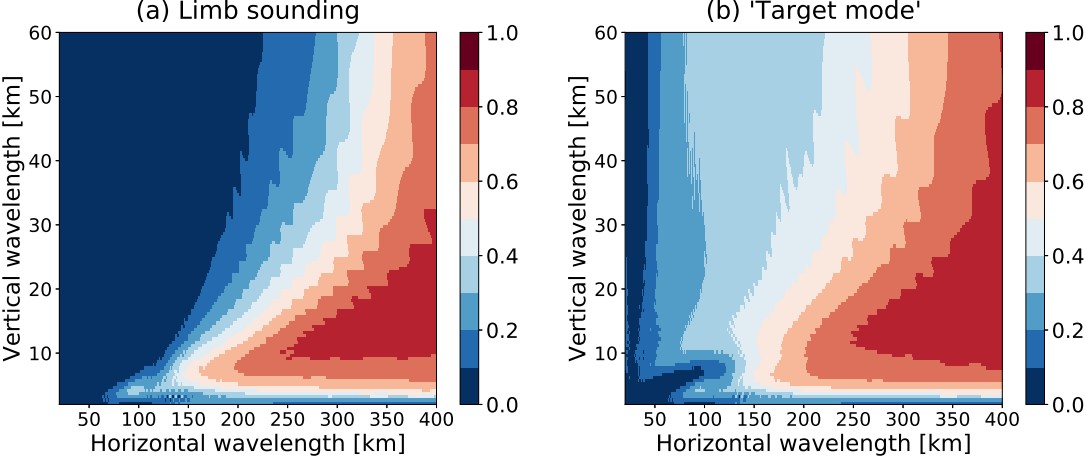

**Figure 8.** Gravity wave sensitivity function. The ratio of retrieved wave amplitude to true wave amplitude as function of horizontal and vertical wavelength is shown. Figure (a) is the sensitivity function for pure limb measurements, whereas Figure (b) is for 'target mode' measurements as specified in Sect. 5.

### 5.3 Sensitivity study

The quality of a gravity wave amplitude-, wavelength- and phase- retrieval depends on the wavelength of the wave. The long LOS of limb observations is ideal for the reconstruction of long horizontal wavelengths, whereas gravity waves with short horizontal wavelengths are likely underestimated or cannot be measured at all. A measure to assess the sensitivity of an observation system to retrieve gravity wave parameters is the so called gravity wave sensitivity function (Preusse et al., 2002). It defines how a wave perturbation of a given horizontal and vertical wavelength is reproduced by a retrieval. One option to determine this gravity wave sensitivity function is to perform the retrieval for the wavelength of interest. However, for a tomographic retrieval this is computationally very expensive. Alternatively, we can derive the gravity wave sensitivity function more efficiently by using the averaging kernel matrix method (Ungermann et al., 2010a). The basic idea of this method is to assume that the forward model can be approximated linearly for a small perturbation as induced by a gravity wave. In this case, the averaging kernel matrix $\mathbf{A}$ would be identical for the unperturbed atmosphere and the perturbed atmosphere. If the unperturbed atmosphere $\boldsymbol{x}_b$ is assumed to be the same as the a priori data $\boldsymbol{x}_a$, the retrieval result $\boldsymbol{x}_f$ and the a priori vector $\boldsymbol{x}_a$ are related as follows:

$$\boldsymbol{x}_f - \boldsymbol{x}_a = (\mathbf{A}(\boldsymbol{x}_a + \boldsymbol{x}_\delta) + (\mathbf{I} - \mathbf{A})\boldsymbol{x}_a) - \boldsymbol{x}_a = \mathbf{A}\boldsymbol{x}_\delta \qquad (17)$$

with $\mathbf{I}$ being the identity matrix and $\boldsymbol{x}_\delta$ being the modulated wave structure. Following this equation, the averaging kernel matrix $\mathbf{A}$ maps the true wave perturbations $\boldsymbol{x}_\delta$ onto the retrieved wave structure $\boldsymbol{x}_f - \boldsymbol{x}_a$. The ratio between $\boldsymbol{x}_f - \boldsymbol{x}_a$ and $\boldsymbol{x}_\delta$ quantifies the sensitivity to reconstruct the gravity wave. Fig. 8 shows the results of sensitivity study adopting this approach for horizontal and vertical wavelengths of 0-400 and 0-60 km, respectively. To illustrate the advancement of the combination





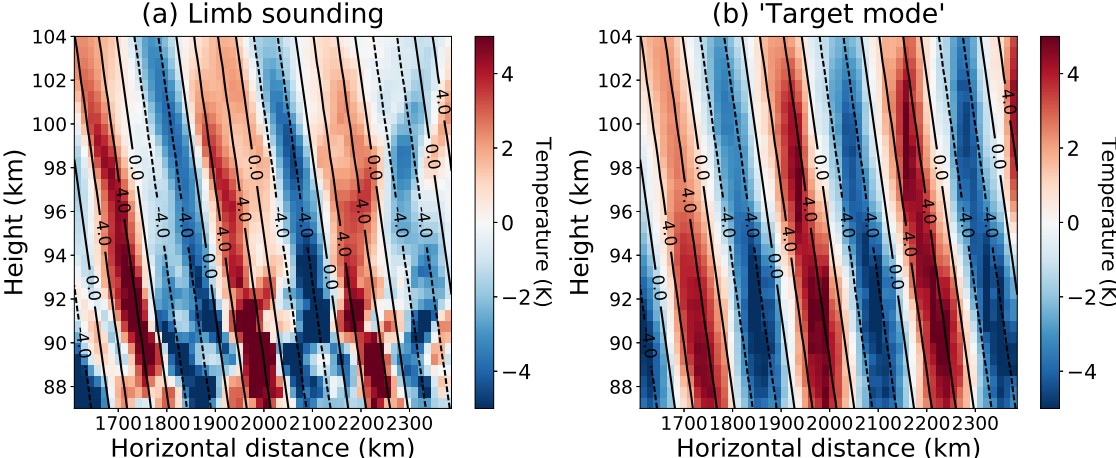

**Figure 9.** Pure limb sounding (a) and 'target mode' (b) retrieval results comparison. The gravity wave has a wavelength of 250 km in horizontal and 40 km in vertical. The climatological background profile was subtracted from the retrieval result. The black lines indicate the true modulated wave structure.

of limb and sub-limb observations for the reconstruction of gravity wave amplitudes, the sensitivity function for a pure limb measurement was also calculated, as shown in Fig. 8 (a). The retrieval setup is the same for both cases, but the additional sub-limb measurements were removed from Fig. 8 (b) to get pure limb simulation in Fig. 8 (a).

For pure limb sounding, GWs with vertical and horizontal wavelengths down to 7 km and 150 km, respectively, can be observed.
The sensitivity to detect short horizontal wavelengths decreases for larger vertical wavelengths, e.g. 250 km at 20 km vertical wavelength, or 325 km at 60 km vertical wavelength, respectively. In Fig. 8 (b), 'target mode' tomography has been performed which considers sub-limb measurements with 15 seconds integration time as well. The GW sensitivity function does not change much for short vertical wavelength compared to pure limb measurements, but gravity waves with large vertical wavelength and short horizontal wavelength become much more visible. For GWs with vertical wavelengths above 15 km, the increase in
horizontal wavelength sensitivity is typically 50-100 km. Another advancement of the 'target mode' is the reduced altitude dependence of the observational filter, as it is also illustrated in Fig. 9 for a GW with 250 km horizontal wavelength and 40 km vertical wavelength. Comparing the retrieved wave structure with the true structure (depicted as the black contour plot), the 'target mode' tomography can reconstruct the wave structure more clearly than the limb mode tomography.

# 6  Conclusion

In recent years, tomographic retrieval approaches have been proposed to reconstruct 2D gravity wave structures. However, the spatial resolution of gravity waves retrieved from this observation mode is limited by the poor horizontal resolution along the instrument's LOS of these instruments.





In this paper, a novel 'target mode' observation combining limb and sub-limb measurements for retrieval of GW parameters in the mesopause region is presented. A tailored retrieval scheme for this observational mode has been presented and its performance has been assessed.

We employed this new approach to simulated measurements of an instrument measuring the $O_2$ A-band nightglow emissions
5   to demonstrate its advantages in resolving 2D atmospheric structures. The retrieval results show that a combination of limb and sub-limb measurements increases the sensitivity to detect short horizontal wavelengths by 50-100 km compared to pure limb sounding. GWs with vertical and horizontal wavelength down to 7 km and 150 km can be resolved. It is shown that its capability of detecting short horizontal wavelength is depending on the vertical wavelength of GWs, e.g. 200 km at 20 km vertical wavelength, and 260 km at 60 km vertical wavelength, respectively.

10  **Acknowledgements**

This work was supported in part by the National Natural Science Foundation of China under Grants 41590852, and the China scholarship council (201404910513).





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
