# Peer review of "Tomographic reconstruction of atmospheric gravity wave parameters from airglow observations"

_Atmospheric Measurement Techniques, 2017_

## Referee Comment (RC1) · Anonymous Referee #1 · 28 Apr 2017

The paper "Tomographic reconstruction..." by R. Song et al. describes a new variant of a tomographic retrieval which is custom tailored to a not-yet existing measurement system planned to be operated in a novel measurement mode. As a methodical pre-flight-study the scope of the paper fits well in AMT but a couple of details should be clarified prior to publication. In general, the paper is well written, well organized, scientifically sound, and as far as I can judge, all relevant literature has been referenced.

Scientific issues:

p2 l30: I am not quite sure if the term "oversampling" is adequate here, at least in the context of MIPAS (although MIPAS is not explicitly mentioned here, the Carlotti et al. reference hints at MIPAS). Von Clarmann et al., Atmos. Meas. Techn., Vol. 2(1), 47-54,

2009, "The horizontal resolution of MIPAS", find that the horizontal resolution of MIPAS in terms of along-track information smearing is better than the horizontal sampling in terms of the horizontal spacing of measurement geo-locations. Thus there is under-sampling, not oversampling. Either reword, or define clearly in which sense you use the admittedly ambiguous term "oversampling". Does it refer to the retrieval space or the measurement space, etc?

p5 l5 and p8 l15: On page 5 the atmosphere is assumed to be so opaque that any signal from the lower atmosphere and surface can be ignored. On page 8 the atmosphere is so optically thin that no self-absorption has to be considered. These two approximations seem to be in conflict with each other, except if the transmission jumps from 1 to 0 at the target air volume. I do not doubt that the approximations made can somehow be justified but I think that a little more discussion is needed to refute the apparent contradiction. Particularly in the sub-limb mode I expect that you either get considerable signal from altitudes below the target volume, or that the atmosphere in front/above the target volume is not really transparent.

p6, general comment: No statement is made how well the measurement geometry is known, and in consequence, how well the pressures at the tangent points are known. The measured signal does not only depend on temperature but also on the number of molecules along the ray-path (or more precisely: the transparent and semi-transparent part of the ray-path). How is this multi-variable problem solved? Or is the tacit assumption made that the actual measurement geometry and pressure distribution are perfectly known? The manuscript should be a bit clearer with respect to this.

Wording and presentation issues and technical corrections:

Abstract: About a third of the abstract are like an introduction. I would prefer an abstract which includes less general introductory information but more methodical information and/or results.

p1 l10: There is no "2-dimensional atmospheric state". Better say "allow for tomographic 2-dimensional reconstruction of the atmospheric state".

p1 l11: "As no real data are available" sounds too defensive to my ears. It is fully legitimate to present pre-flight studies and retrieval sensitivity studies in AMT. Why not simply "The feasibility of this tomographic retrieval approach is assessed using simulated measurements".

p1 l12: "much smaller" than what?

p1 l20: comma after "(GWs)"

p2 l14 "they include": not quite clear what "they" refers to. I suggest to reword.

p3 l2 "limited-angle tomography". This sounds as if it was a technical term but I have never heard it before. Please either define this technical term, or avoid it and use generic terms instead.

p3 l3/4: The technical term "target mode" is used here twice but defined only in line 18. I suggest to write "In Section 2 we present the observation strategy which we call 'target mode' observations. Then it is clear that you give this name to something new and the reader will not wonder if he/she has missed anything.

p3 l 16: Please define the term "sub-limb sounding".

p3 l 23: The description of the geometry is somewhat unclear. You talk about "limb sounding", not "limb imaging". This implies that during one limb scan the viewing geometry is changing all the time. It is thus not clear how multiple consecutive profiles can be obtained while a limb view is kept. The text would be much clearer if you distinguish between limb scanning (usually used as a synonym to limb sounding) or limb imaging (recording of multiple ray-paths at the same time) is used. The statement "The instrument will keep the limb view" currently has three possible different meanings: 1. A series of measurements is made using the SAME tangent altitude. 2. A profile of limb radiances is measured simultaneously with a 1D imaging device. 3. Tangent altitudes change while the limb is scanned, and the statement is just meant to tell me that

it is not switched from the limb-scanning mode to the sub-limb mode. Please clarify. Perhaps use a weaker wording than "will keep this limb view"; perhaps say "The instrument will continue to measure under limb geometry for a period of time"; and finally clarify what type of "vertival profiles" are measured. I guess "vertical radiance profiles".

p5 l18: Is A1 really a "transmission probability" or do you mean a "transition probability"

p6 l2: Is the A1 here the same as in Eq 3: Please use different symbols for different designates, and/or use the same technical terms for the same designates.

p8 Eq 13: Since Sa-1 is not the inverse of a covariance matrix but a freely defined regularization term, I find it inadequate to use the symbol Sa here, which is usually applied only for probabilistic a priori covariance matrices.

p9 l2: Not clear what "It" refers to. The content suggests that it refers to the entire regularization procedure but grammatically it refers to the a priori data alone, which does NOT ensure that a unique solution can be obtained. I suggest: "The second term in the cost function (Eq. 13) ensures that..."

p9 l5: references to Levenberg, Marquardt, and the implementation actually used would be appropriate.

p9 l11: you may wish to add the term "unstable solution" here

p10 Eq 16: If you used the convention that f' = K, then the equation would be easier to recognize (and, of course, define K prior to its use).

p10 l20: For the non-specialist it would be helpful to clarify if you perturb the temperature field only or if you adjust pressure (and with this also absolute concentrations of species) hydrostatically.

p14 l17: Has the acronym LOS already been defined? With respect to the "poor horizontal resolution", see my comment on p2 l30: At least for MIPAS it is the horizontal sampling and not the resolution of the measurement itself which is limiting the horizon-

tal resolution of the data product.

---

## Referee Comment (RC2) · Anonymous Referee #2 · 21 Jul 2017

Review comments for "tomographic reconstruction of atmospheric gravity wave parameters from airglow observations" by Song et al.

This manuscript thoroughly describes a methodology of retrieving 3D gravity wave parameters (wavelength and amplitude) from a synthetic remote sensing instrument that is designed to work on the "target mode" at O2 A-band. Taking the advantage of combining both "limb" and "sub-limb" strengths, this "target mode" can capture the majority of the gravity waves on the spectrum except the very small ones (both horizontal and vertical wavelengths are small). The aiming region is at mesopause where a lot of gravity wave breaking and secondary generation occur, so this methodology, together

with the specially designed viewing geometry, is likely a powerful tool of investigating the gravity wave dynamics, and mesosphere-thermosphere coupling on a global scale.

This paper is well-written. The flow is smooth, the logic is strict, and the presentation is concise and clear. It well suits the journal of AMT, and deserves a final publication.

I have some broad questions and comments that I hope the authors can address before final publication. I don't want to hit the "major revision" button because the following comments are indeed not too critical. But I sincerely hope the authors could take at least #3 seriously and add one figure to address this issue.

1. Although the "observation" is synthetic, the paper is not clear about what the designed orbit, scan frequency, global coverage, etc. should be, so readers have no idea whether this "mission", if successfully launched, could be suitable for case studies and climate studies.

2. Similar to the above question, the integration time of each limb/sub-limb view seems to impact the sensitivity window (i.e., Fig. 8). Other than gain (or signal-noise ratio), I don't see a clear way that they are connected. Can you quantitatively elaborate why?

3. The authors mentioned that one of the difficulty this "target mode" can conquer is that we don't need two adjacent orbits to determine the horizontal wavelength. But in my understanding, the aliasing effect still exists, i.e., the satellite instrument is still only sensitive to wave fronts that are parallel to the LOS. In the "pseudo-retrieval", the input "truth" is also a linear gravity wave with wave front parallel to the LOS. What about other direction? I think an evaluation of the dependence of retrieved wave parameter as a function of wave vector direction is necessary to show to the readers. In addition, it would be nice to briefly discuss the situation of a mixture of two linear waves, and other types of GWs, e.g., circular rings. The general interests lie in the fact that many GWs become non-linear at the mesopause.

4. Regarding the horizontal wavelength, there is still no way to decompose it to

lumbda_x and lumbda_y, is that right?

Minor points: P1, L3: wind system -> wind structure. P1, L15: for -> from P2, L8: they include -> these datasets include P2, L10: In Wu and Waters (1996), they used the saturated radiance (and hence, it's sub-limb technique, not limb, read Wu and Eckermann (2008, JAS) for details), not the retrieved temperature. P2, L15: Please include Gong et al. [2012, ACP] and Hoffmann et al. [2016, ACP] in the reference list. P2, L23: short horizontal waves –> waves with short horizontal wavelengths. P2, L32: add "small" before "structure". P3, L19: observation -> observations.

Fig. 5: My understanding is that this figure shows the weighting function of each channel, correct? If that's the case, I think it's better to draw the weighting function line as a function of altitude for each channel, using different color to represent different channels would be a better idea. Right now it's not straightforward of the subtle difference of weighting function peak at different altitude.
* * *

---

## Author Comment (AC1) · 26 Aug 2017

**Responses to reviewers' comments on**
"Tomographic reconstruction of atmospheric gravity wave parameters from airglow observations" by Song et al.

The authors would like to thank the reviewers for their valuable comments, which helped us to improve the quality of this manuscript. We have addressed all the comments, and the reply to each comment is highlighted in blue as follows.

**Reply to Anonymous Referee #1**

The paper "Tomographic reconstruction..." by R. Song et al. describes a new variant of a tomographic retrieval which is custom tailored to a not-yet existing measurement system planned to be operated in a novel measurement mode. As a methodical preflight-study the scope of the paper fits well in AMT but a couple of details should be clarified prior to publication. In general, the paper is well written, well organized, scientifically sound, and as far as I can judge, all relevant literature has been referenced.

We thank the referee for carefully reviewing the manuscript and for the positive comments.

Scientific issues:

p2 l30: I am not quite sure if the term "oversampling" is adequate here, at least in the context of MIPAS (although MIPAS is not explicitly mentioned here, the Carlotti et al. reference hints at MIPAS). Von Clarmann et al., Atmos. Meas. Techn., Vol. 2(1), 47-54, 2009, "The horizontal resolution of MIPAS", find that the horizontal resolution of MIPAS in terms of along-track information smearing is better than the horizontal sampling in terms of the horizontal spacing of measurement geo-locations. Thus there is undersampling, not oversampling. Either reword, or define clearly in which sense you use the admittedly ambiguous term "oversampling". Does it refer to the retrieval space or the measurement space, etc?

Thanks for the suggestion. We agreed that the term "oversampling" is inadequate here. The original meaning of this sentence is to show that better horizontal resolution can be achieved for some limb sounders, such as MLS and MIPAS. As such instruments observe the atmosphere along the track of the orbit, the atmospheric variability along the LOS can be considered in the retrieval. In the paper "The

horizontal resolution of MIPAS" by Von Clarmann et al., the authors explained clearly that the along-track smearing is two times smaller than the horizontal sampling, which means the atmosphere is undersampled. Therefore, we have rewritten this sentence and added the reference of Von Clarmann et al. "...mitigate this general limitation by considering the horizontal variability of the atmosphere in the retrieval (Livesey and Read, 2000; Carlotti et al., 2001; von Clarmann et al.,2009)".

p5 l5 and p8 l15: On page 5 the atmosphere is assumed to be so opaque that any signal from the lower atmosphere and surface can be ignored. On page 8 the atmosphere is so optically thin that no self-absorption has to be considered. These two approximations seem to be in conflict with each other, except if the transmission jumps from 1 to 0 at the target air volume. I do not doubt that the approximations made can somehow be justified but I think that a little more discussion is needed to refute the apparent contradiction. Particularly in the sub-limb mode I expect that you either get considerable signal from altitudes below the target volume, or that the atmosphere in front/above the target volume is not really transparent.

In the infrared, this model assumes the atmosphere to be optically thick in the lower atmosphere and optically thin in the upper atmosphere. To avoid the conflict in the manuscript, we added a sentence to explain the assumption prior to use: "In the infrared the lower atmosphere is optically thick, whereas the upper atmosphere can be considered as optically thin. Since the $O_2$ A-Band is a transition to the $O_2$ ground state, the atmosphere becomes optically thick at stratopause altitudes. Therefore, any emission from the Earth's surface or tropospheric altitudes cannot reach the upper mesosphere at these wavelengths. At nightglow layer altitudes (upper mesosphere / lower thermosphere) the atmosphere is optically thin for the wavelengths considered. In our case,...".

p6, general comment: No statement is made how well the measurement geometry is known, and in consequence, how well the pressures at the tangent points are known. The measured signal does not only depend on temperature but also on the number of molecules along the ray-path (or more precisely: the transparent and semi-transparent part of the ray-path). How is this multi-variable problem solved? Or is the tacit assumption made that the actual measurement geometry and pressure distribution are perfectly known? The manuscript should be a bit clearer with respect to this.

Thanks for the suggestion. In this simulation, the pressure is not needed and the actual measurement geometry is assumed perfectly known. The atmospheric temperature $T$ is retrieved on the tangent altitudes in this study.

Wording and presentation issues and technical corrections:

Abstract: About a third of the abstract are like an introduction. I would prefer an abstract which includes less general introductory information but more methodical information and or results.

Thanks. We have revised the abstract according to your suggestion.

p1 l10: There is no "2-dimensional atmospheric state". Better say "allow for to-mographic 2-dimensional reconstruction of the atmospheric state".

Agreed. This change has been made.

p1 l11: "As no real data are available" sounds too defensive to my ears. It is fully legitimate to present pre-flight studies and retrieval sensitivity studies in AMT. Why not simply "The feasibility of this tomographic retrieval approach is assessed using simulated measurements".

[Figure]

Thanks for the suggestion. This sentence has been rewritten.

p1 l12: "much smaller" than what?

The text has been revised to clarify this. "It shows that one major advantage of this observation strategy is that GWs can be observed on a much smaller scale than conventional observations."

p1 l20: comma after "(GWs)"

A comma has been added after "(GWs)".

p2 l14 "they include": not quite clear what "they" refers to. I suggest to reword.

This sentence has been reworded. "GWs can be also characterized by nadir viewing instruments, such as the Atmospheric Infrared Sounder (AIRS) (Alexander and Barnet, 2007; Hoffmann and Alexander, 2009; Ern et al., 2017) and the Advanced Microwave Sounding Unit (AMSU) (Wu, 2004)".

p3 l2 "limited-angle tomography". This sounds as if it was a technical term but I have never heard it before. Please either define this technical term, or avoid it and use generic terms instead.

This sentence has been rewritten to clarify this point. "This results in multi-angle observations of the target volume, such that a tailored retrieval scheme can be applied. This differs from classical limited-angle tomography, where only observations within a limited angular range are taken for the reconstruction."

p3 l3/4: The technical term "target mode" is used here twice but defined only in line 18. I suggest to write "In Section 2 we present the observation strategy which we call 'target mode' observations. Then it is clear that you give this name to something new and the reader will not wonder if he/she has missed anything.

Thanks for the suggestion. This sentence has been revised.

p3 l 16: Please define the term "sub-limb sounding".

The term "sub-limb" has been defined in text. "The sub-limb sounding has a similar geometry with limb sounding, whereas the tangent heights are near or below the surface."

p3 l 23: The description of the geometry is somewhat unclear. You talk about "limb sounding", not "limb imaging". This implies that during one limb scan the viewing geometry is changing all the time. It is thus not clear how multiple consecutive profiles can be obtained while a limb view is kept. The text would be much clearer if you distinguish between limb scanning (usually used as a synonym to limb sounding) or limb imaging (recording of multiple ray-paths at the same time) is used. The statement "The instrument will keep the limb view" currently has three possible different meanings: 1. A series of measurements is made using the SAME tangent altitude. 2. A profile of limb radiances is measured simultaneously with a 1D imaging device. 3. Tangent altitudes change while the limb is scanned, and the statement is just meant to tell me that it is not switched from the limb-scanning mode to the sub-limb mode. Please clarify. Perhaps use a weaker wording than "will keep this limb view"; perhaps say "The instrument will continue to measure under limb geometry for a period of time"; and finally clarify what type of "vertical profiles" are measured. I guess "vertical radiance profiles".

[Figure]

Thanks for finding and explaining this point that was not clear in the manuscript. We have revised this paragraph to avoid misleading the readers. First, the term "limb sounding" has been replaced by "limb imaging". Second, we accepted your suggestion for revising the sentence "The instrument will keep this limb view...". It has been rewritten as "The instrument will continue to measure under limb geometry for a period of time, and multiple consecutive vertical radiance profiles will be taken during this time".

p5 l18: Is A1 really a "transmission probability" or do you mean a "transition probability".

Corrected as "transition probability".

p6 l2: Is the A1 here the same as in Eq 3: Please use different symbols for different designates, and/or use the same technical terms for the same designates.

In Eq 3, the A-band transition probability is represented by $A_1$. In Eq 6, the wave amplitude is represented by $A$.

p8 Eq 13: Since $\mathbf{S}_a^{-1}$ is not the inverse of a covariance matrix but a freely defined regularization term, I find it inadequate to use the symbol Sa here, which is usually applied only for probabilistic a priori covariance matrices.

Thanks for the suggestion. Therefore, we use $\mathbf{R}$ instead of $\mathbf{S}_a^{-1}$ in Eq 13 and the following text.

p9 l2: Not clear what "It" refers to. The content suggests that it refers to the entire regularization procedure but grammatically it refers to the a priori data alone, which does NOT ensure that a unique solution can be obtained. I suggest: "The second term

in the cost function (Eq. 13) ensures that..."

Agreed. This sentence has been revised to : "The second term in the cost function (Eq. (13)) ensures that..."

p9 l5: references to Levenberg, Marquardt, and the implementation actually used would be appropriate.

Relevant references have been added. "(Levenberg, 1944; Marquardt, 1963; Ceccherini and Ridolfi, 2010)".

p9 l11: you may wish to add the term "unstable solution" here

Accepted. Sentence has been revised as "... while a small value will give an unstable solution".

p10 Eq 16: If you used the convention that $f' = \mathbf{K}$, then the equation would be easier to recognize (and, of course, define K prior to its use).

Agreed. $f'$ has been replaced by $\mathbf{K}$ for easier recognition. A sentence has been added to define $\mathbf{K}$: "where $\mathbf{K}$ is the Jacobian of the forward model $f$ at atmospheric state $x$".

p10 l20: For the non-specialist it would be helpful to clarify if you perturb the temperature field only or if you adjust pressure (and with this also absolute concentrations of species) hydrostatically.

A sentence has been added to clarify this: "The temperature, atmospheric density and concentrations of various constituents are perturbed by this simulated wave."

p14 l17: Has the acronym LOS already been defined? With respect to the "poor horizontal resolution", see my comment on p2 l30: At least for MIPAS it is the horizontal sampling and not the resolution of the measurement itself which is limiting the horizontal resolution of the data product.

The acronym LOS is defined in p3 l30 (discussion paper): "Figure. 2 shows how the line-of-sights (LOSs) of...". With respect of the "poor horizontal resolution", this point has been revised in the introduction according to the comment on p2 l30.

---

## Author Comment (AC2) · 26 Aug 2017

[amtd, manuscript]copernicus [parfill]parskip

**Responses to reviewers' comments on**
**"Tomographic reconstruction of atmospheric gravity wave parameters from airglow observations" by Song et al.**

The authors would like to thank the reviewers for their valuable comments, which helped us to improve the quality of this manuscript. We have addressed all the comments, and the reply to each comment is highlighted in blue as follows.

**Reply to Anonymous Referee #2**

This manuscript thoroughly describes a methodology of retrieving 3D gravity wave parameters (wavelength and amplitude) from a synthetic remote sensing instrument that is designed to work on the "target mode" at O2 A-band. Taking the advantage of combining both "limb" and "sub-limb" strengths, this "target mode" can capture the majority of the gravity waves on the spectrum except the very small ones (both horizontal and vertical wavelengths are small). The aiming region is at mesopause where a lot of gravity wave breaking and secondary generation occur, so this methodology, together with the specially designed viewing geometry, is likely a powerful tool of investigating the gravity wave dynamics, and mesosphere-thermosphere coupling on a global scale. This paper is well-written. The flow is smooth, the logic is strict, and the presentation is concise and clear. It well suits the journal of AMT, and deserves a final publication. I have some broad questions and comments that I hope the authors can address before final publication. I don't want to hit the "major revision" button because the following comments are indeed not too critical. But I sincerely hope the authors could take at least #3 seriously and add one figure to address this issue.

We thank the reviewer for providing a thorough review and offering valuable suggestions. We have revised the manuscript according to the comments. Especially for #3, an additional figure has been used to clarify the problem.

1. Although the "observation" is synthetic, the paper is not clear about what the designed orbit, scan frequency, global coverage, etc. should be, so readers have no idea whether this "mission", if successfully launched, could be suitable for case studies and climate studies

Thanks for the suggestion. A few sentences have been added in the beginning of Sect 5.2 to clarify this:
" The satellite platform is simulated in an approximately 600 km sun-synchronous orbit with an inclination angle of 98°. The instrument will employ a 2D detector array consisting of about 40 × 400 super pixels. It measures in the spectral regions from 13082 to 13103 cm[1] within the altitude range from ≈60 to 120 km in limb imaging measurements.".

2. Similar to the above question, the integration time of each limb/sub-limb view seems to impact the sensitivity window (i.e., Fig. 8). Other than gain (or signal-noise ratio), I don't see a clear way that they are connected. Can you quantitatively elaborate why?

Yes, the integration time will affect the sensitivity window as shown in Fig. 8. The sensitivity to horizontal waves can be increased by using shorter integration time. The shorter the integration time is, the less horizontal information of the atmosphere will be smoothed in each limb or sub-limb view. However, this integration time should also be adequate such that enough photons can be received by the instrument. In this experiment we aim to show the readers that, for the same instrument the horizontal resolution can be improved by incorporating sub-limb measurements, even if the integration time of the instrument itself can not be improved. For this reason, we didn't show the influence of integration time on the sensitivity window in Fig. 8. Then it's clear to see how the 'target mode' can improve the performance in analyzing horizontal wavelength of the waves.

3. The authors mentioned that one of the difficulty this "target mode" can conquer is that we don't need two adjacent orbits to determine the horizontal wavelength. But in my understanding, the aliasing effect still exists, i.e., the satellite instrument is still only sensitive to wave fronts that are parallel to the LOS. In the "pseudo-retrieval",

the input "truth" is also a linear gravity wave with wave front parallel to the LOS. What about other direction? I think an evaluation of the dependence of retrieved wave parameter as a function of wave vector direction is necessary to show to the readers. In addition, it would be nice to briefly discuss the situation of a mixture of two linear waves, and other types of GWs, e.g., circular rings. The general interests lie in the fact that many GWs become non-linear at the mesopause.

Agreed. In this 'target mode', the instrument is sensitive to the wave vector that has a component parallel to the LOS. This effect was not clarified in the manuscript. We added a figure to illustrate the viewing geometry between the LOS and wave vector. A few sentences were used to clarify this effect before the results of the sensitivity study are presented:
"For this 'target mode', the observed horizontal wavelength is the wavelength projected along the LOS. In general, there is an angle $\alpha$ between the LOS and the horizontal wave vector. Therefore, the observed horizontal wavelength $\lambda_x$ is a factor of $1/(\cos\alpha)$ larger than the real horizontal wavelength $\lambda_r$, as illustrated in the added figure. In this sensitivity study, the horizontal wavelength discussed is the observed horizontal wavelength $\lambda_x$".
The focus of this paper is to show how well the horizontal resolution can be improved by performing the 'target mode' observation. In the case of 2-dimensional retrieval, a combination of vertical and horizontal information is enough for the analysis of any 2-d waves. We are currently working on another retrieval strategy to get more information on the orientation of the wave vector. However, this topic is rather complex and would fill another paper.

4. Regarding the horizontal wavelength, there is still no way to decompose it to lambda_x and lambda_y, is that right?

Yes, right. This point has been clarified in Comment #3.

Minor points: P1, L3: wind system -> wind structure. P1, L15: for -> from P2, L8: they include -> these datasets include P2, L10: In Wu and Waters (1996), they used the saturated radiance (and hence, it's sub-limb technique, not limb, read Wu and Eckermann (2008, JAS) for details), not the retrieved temperature. P2, L15: Please include Gong et al. [2012, ACP] and Hoffmann et al. [2016, ACP] in the reference list. P2, L23: short horizontal waves –> waves with short horizontal wavelengths. P2, L32: add "small" before "structure". P3, L19: observation -> observations.

Thanks for the detailed reading. All these minor points have been addressed in the revised manuscript.

Fig. 5: My understanding is that this figure shows the weighting function of each channel, correct? If that's the case, I think it's better to draw the weighting function line as a function of altitude for each channel, using different color to represent different channels would be a better idea. Right now it's not straightforward of the subtle difference of weighting function peak at different altitude.

This figure is not the weighting function for each channel. It shows the simulated radiance at different tangent heights.